# The Effect of Smartphone-Based Cognitive Training on the Functional/Cognitive Markers of Schizophrenia: A One-Year Randomized Study

**DOI:** 10.3390/jcm9113681

**Published:** 2020-11-16

**Authors:** Marek Krzystanek, Krzysztof Krysta, Mariusz Borkowski, Katarzyna Skałacka, Jacek Przybyło, Artur Pałasz, Davor Mucic, Ewa Martyniak, Napoleon Waszkiewicz

**Affiliations:** 1Clinic of Psychiatric Rehabilitation, Department of Psychiatry and Psychotherapy, Faculty of Medical Sciences, Medical University of Silesia in Katowice, Ziołowa 45/47, 40-635 Katowice, Poland; krzystanekmarek@gmail.com (M.K.); mariusz.amanogawa@gmail.com (M.B.); evamartyniak@gmail.com (E.M.); 2Institute of Psychology, University of Opole, Kopernika 11A Street, 45-040 Opole, Poland; katarzyna.skalacka@uni.opole.pl; 3Multispecialistic Voivodship Medical Clinic in Katowice, Lompy 16, 40-038 Katowice, Poland; jprzybylo@gmail.com; 4Department of Histology, Faculty of Medical Sciences, Medical University of Silesia in Katowice, Medyków 18, 40-752 Katowice, Poland; artiassone@gmail.com; 5The Little Prince Treatment Centre, Havneholmen 82, 5th, V 1561 Copenhagen, Denmark; dmucic@gmail.com; 6Department of Psychiatry, Medical University of Białystok, Plac Brodowicza 1 Str., 16-070 Choroszcz, Poland; napoleon.waszkiewicz@umb.edu.pl

**Keywords:** paranoid schizophrenia, cognitive impairment, cognitive training, cognitive markers, functional markers, smartphone application, telephone-based intervention, cognitive remediation therapy

## Abstract

Background: Cognitive impairment is associated with long-term disability that results in the deterioration of both the social and professional status of individuals with schizophrenia. The impact of antipsychotic therapy on cognitive function is insufficient. Cognitive training is therefore proposed as a tool for cognitive rehabilitation in schizophrenia. In this study we investigated the effect of self-administered cognitive training using a smartphone-based application on the cognitive function of paranoid schizophrenia patients focusing on response time, correct answer rate, incorrect answer rate, and fatigability to check, if these functions can be functional markers of successful cognitive-smartphone rehabilitation. Methods: 1-year multicenter, open-label randomized study was conducted on 290 patients in a state of symptomatic remission. 191 patients were equipped with the full version of the application and conducted cognitive training twice a week. Reference group (*n* = 99) was provided with a version of the application having only limited functionality, testing the cognitive performance of patients every 6 months. Results: Statistically significant improvement was observed in both the rate of correct answers (by 4.8%, *p* = 0.0001), and cognitive fatigability (by 2.9%, *p* = 0.0001) in the study group, along with a slight improvement in the rate of incorrect answers (by 0.9%, *p* = 0.15). In contrast, the reference group, who performed cognitive training every 6 months, demonstrated no significant changes in any cognitive activities. Conclusions: Cognitive trainings facilitated by a smartphone-based application, performed regularly for a longer period of time are feasible and may have the potential to improve the cognitive functioning of individuals with schizophrenia. Correct answers and cognitive fatigability have potential to be functional markers of successful smartphone-based psychiatric rehabilitations in schizophrenia patients.

## 1. Introduction

Cognitive impairment is recognized as a key manifestation of schizophrenia, and is associated with long-term disability that deteriorates both the social and professional status of affected individuals [1,2]. In contrast to positive and negative symptoms, which escalate and diminish in parallel with the course of the disease [3], cognitive functions tend to remain constant during periods of psychosis and remission [4,5]. Therefore, cognitive deficits can be considered to be primary manifestations of schizophrenia, which appear earlier than other symptoms of the disease [6].

Cognitive impairment results from deficits in attention and working memory, stemming from difficulties with holding particular elements in short-term memory [7]. Cognitive deficits also include problems with learning, psychomotor speed and executive functions such as abstract thinking and problem-solving [8]. Apart from negative symptoms, it is cognitive dysfunction that is considered to be responsible for significant functional impairment of individuals affected by schizophrenia [9,10,11].

As many as 98% patients with schizophrenia exhibit lower cognitive function than may be expected based on premorbid functioning and IQ estimates, as well as parental education [10]. In IQ tests, individuals with schizophrenia score approximately one standard deviation (or 15 IQ points) below the general population mean [11]. Additionally, cognitive impairment includes significant worsening of visual and verbal memory, attention, reasoning, executive function, processing speed, spatial abilities, and abstract thinking [12,13,14]. Persistence of cognitive dysfunction represents the worst prognostic factor for patients’ social functioning and work performance [15,16].

Some of the cognitive disturbances such as working memory, executive function, and attention are included in the concept of endophenotype that is closely related to schizophrenia marker. These cognitive functional markers result in susceptibility to schizophrenia, are repeatable in the family of ill patients, so are partly hereditary, and determine the susceptibility to falling ill [17].

Remission of positive symptoms remains an important goal in schizophrenia treatment; however, it is associated with only a 5% improvement in the quality of life in affected individuals [18]. Most studies indicate that persistent functional disability in schizophrenia is due to cognitive impairment as well as negative symptoms as opposed to positive symptoms [11]. First- and second-generation antipsychotic drugs in current use have only a slight effect on cognitive disorders in patients with schizophrenia [16,19,20]. A large meta-analysis including 34 studies revealed that conventional neuroleptic agents exert modest to moderate benefit in terms of cognitive deficits [21].

A non-pharmacological attempt to improve cognitive functions is cognitive rehabilitation therapy (CRT). According to the CRT Experts Workshop in 2010, it is defined as a behavioral intervention to improve cognitive processes such as attention, memory, executive function, social cognition, and metacognition that remains stable in time and affects day-to-day functioning [22]. Among neurocognitive training, two distinct classes are recognized: CRT, consisting of cognitive tasks to be performed repeatedly, and compensatory cognitive training, consisting of interventions designed to compensate for patient-specific cognitive deficits [23].

So far, the efficacy of cognitive training exercises to improve cognitive functions in individuals suffering from schizophrenia is considered to be slight to moderate. Nevertheless, many studies revealed its positive impact on patients’ well-being, especially in the domains of cognition, ability to work, social cognition, independent living skills, and social adjustment [22,24,25,26]. Given the prominent decrease in quality of life in individuals with schizophrenia due to cognitive impairment along with the insufficient efficacy of neuroleptic agents in normalizing cognitive function, the cognitive training represents an important strategy in the management of schizophrenia. One of the forms of CRT is remote training in the form of a mobile application on a smartphone.

Telemedical solutions are becoming increasingly significant in most branches of modern medicine, including psychiatry. According to recent studies, both ownership and usage of mobile phones among people with schizophrenia exceeds 80%, and is still growing [27,28]. More importantly, individuals with schizophrenia are willing to use smartphone-based telemedical tools [29], and they exhibit high rates of compliance (up to 94%) with mobile-based treatment [27]. Additionally, mobile technologies are recognized as safe and well-accepted by the patients [27,30].

Availability of smartphone-based applications oriented toward the management of schizophrenia is growing [27,31]. However, the use of mobile modalities such as smartphones or tablets to conduct the cognitive training in individuals with schizophrenia is poorly documented. Dang et al. (2014) described a tablet-based intervention on 17 first-episode schizophrenia patients [32]. For cognitive training, patients were asked to spend one hour per day, five days per week playing a user-friendly game available on an iPad. This intervention led to a significant improvement in patients’ working memory [32]. In another preliminary study, Biagianti et al. (2017) ran cognitive training exercises on a BrainHQ platform, and asked each patient to complete 40 h of the training [33]. Their study demonstrated that cognitive training exercises conducted on tablets are comparable in terms of both feasibility and efficacy to those conducted on desktop computers [33]. Both of the above studies, although preliminary, indicate a profound potential of remote devices in cognitive rehabilitation of patients with schizophrenia, especially by facilitating access to mental health services.

Because of the minor effect of antipsychotic treatment on cognitive impairment, cognitive training exercises are being proposed as an alternative in management of schizophrenia. The aim of the study was to check whether it is possible to use high level of difficulty short sessions of cognitive training (CT), conducted on a smartphone regularly for a long period of time in improving cognitive functions in individuals with schizophrenia. Although endophenotype-related cognitive functions are generally stable disease markers, we planned the study to check if some of the cognitive functions as response time, correct answer rate, incorrect answer rate, and fatigability can be improved by smartphone-based cognitive training in schizophrenia patients, as well as if these functions have potential to be markers of successful smartphone-based rehabilitation.

## 2. Materials and Methods

### 2.1. Study Design and Participants

This multicenter, open-label, randomized trial included a total of 290 outpatient patients with paranoid schizophrenia; 199 constituted the study group, and 91, the reference group. The reference group was intended to reflect patients who were outpatient too, and did not undergo CT trainings. The disproportion between the study group and the reference group was justified, first of all, by the need to gather a large study group to obtain significant results, on the other hand, we predicted a large number of patients who drop-out of the study. When designing the study, we assumed a 50% drop-out level of patients in the study group.

The patients were recruited from 27 centers in Poland. All patients had received a diagnosis of paranoid schizophrenia according to common ICD-10 criteria within the past 10 years prior to the survey. All patients were in symptomatic remission (mild symptoms not affecting daily functioning and behavior), and their schizophrenia symptoms were stable at a mild level (enabling daily functioning) for at least 6 months prior to study enrolment [34]. PANSS was used in order to determine severity of the symptoms. All study participants had constant access to a high-speed (3G) Internet connection. Exclusion criteria established were: A co-existing psychiatric condition (particularly schizophrenia-like syndromes or organic psychotic disorders); an unstable mental (acute episodes in the past 6 months) or physical state (serious or chronic somatic disease); inability to use an electronic device with a touch screen; pregnancy or lactation; participation in another clinical trial in the past 6 months, or any other reason that, according to the investigator, prevented the individual from participating in a clinical study.

Patients were recruited between January and July of 2014. All patients provided written informed consent to participate in the study. After patient registration, the MONEO system performed randomization of patients either to the study group (who received a full version of the MONEO application) or to the reference group (who received an inactive version of the application with limited functionality). Figure 1 shows the flow of patients throughout the study. Follow-up ended in November 2015. The study protocol was approved by the Bioethics Committee of the Medical University of Silesia in Katowice No. NN-6501-129/05.

### 2.2. Intervention and Implementation

After enrolment, each patient received a smartphone Samsung Galaxy 3 with the MONEO telemedicine application installed. The software in the study group enabled the patient to conduct cognitive training twice a week. The patient received a reminder about scheduled training 1 day before the training and completed it of his/her own free will. Additionally, the software reminded the patient to take medication 1 h before and after the scheduled time. Our investigators monitored the mental state of the patients using psychometric scales every month via videoconference. We arranged outpatient clinic visits via the application once every 3 months. Detailed description of MONEO application functionality is provided by Krzystanek et al. (2018) [35].

The reference group received a version of the software with functionality limited to three cognitive trainings—at the beginning of the trial and at the 6- and 12-months mark, allowing them to measure the parameters of assessed cognitive parameters.

Cognitive training exercises provided by the application were of a visual character and belong to Cognitive Remediation Therapy class of cognitive training. Each training included three series of exercises, with increasing levels of difficulty. Initially, the patient selected one pictogram from the following: balloons (different sizes and colors), fruits (different kinds and colors), airplanes (different shapes and colors), or flowers (different kinds and colors). The patient received visual instruction to remember the displayed pattern. In the first series, the pattern consisting of two images (representing selected pictogram) which differed in color, was displayed on the screen, and remained visible for 30 s (learning phase). Next, the patient was provided with instructions (visible on the screen for 5 s) to click the green box each time he/she noticed the pattern from the learning phase. Each time the patient clicked appropriately, the reaction was deemed to be correct. In the main phase of the training, patterns of two images with different combinations of colors appeared every 0.8 s; this phase lasted for 2.5 min. Of the combinations displayed in the main phase of the training, 20% constituted the correct one (i.e., as had been displayed in the learning phase). The second and third series were identical to the first one, but the pattern consisted of three (in the second series) or four (in the third series) images representing different colors. Each series began immediately after the previous one had finished. The training ended after the completion of the third series. The MONEO application was therefore able to simultaneously conduct cognitive training exercises and assess patients’ cognitive function by monitoring their reaction time, and their correct and incorrect answers.

### 2.3. Outcomes and Measures

Cognitive functions were assessed based on the following parameters: response time, rate of correct and incorrect answers, as well as lack of reaction. All were measured by the MONEO application during cognitive training exercises. The measurements were done every time the patient conducted cognitive training and were collected separately for each of three series.

Response time was the time elapsed from the display of the correct pattern to the patient’s reaction and was expressed in milliseconds.

The rate of correct answers was defined as the percentage of the number of correct pattern displays to which the patient reacted in relation to the number of all correct pattern displays.

The rate of incorrect answers was defined as the percentage of the number of incorrect pattern displays to which the patient reacted in relation to the number of all correct pattern displays.

The rate of lack of reaction was defined as the percentage of number of correct pattern displays to which the patient did not react in relation to the number of all correct pattern displays. In order to classify a given result as a lack of reaction indicating fatigability as opposed to, for instance, a lack of cooperation, only a lack of reaction preceded by at least 10% of the responses in any series of tasks (irrespective of their correctness) was considered to be a measure of cognitive fatigability.

### 2.4. Statistical Analysis

All patients who completed the scheduled procedures throughout the study were included in the final analysis. Analysis was performed on all cases of cognitive training, regardless of its status (completed or not). Missing data were omitted. Continuous variables were expressed as the number of non-missing observations, arithmetic mean, standard deviation, median, quartiles, and range; categorical variables were expressed as the number of non-missing observations and percentages. Univariate and multivariate statistical tests with repeated measurement for dependent groups were applied (Student’s *t*-test, Wilcoxon test, ANOVA, Welch correction was used if necessary). For multivariate analysis measurements, post-hoc tests were applied (Tukey test, LSD test, Scheffe test, Games-Howell test). The Bonferroni correction was used to counteract the problem of multiple comparisons [36]. All tests were two-tailed. A *p* value ≤ 0.05 was considered statistically significant. All statistical analyses were done using STATISTICA 10 (StatSoft, TIBCO Software, Palo Alto, CA, USA) software.

## 3. Results

### 3.1. Baseline Characteristics of the Patients

We intended to enroll 300 patients in the study; however, 10 initially recruited patients were excluded because of lack of signed informed consent (*n* = 9) or lack of signed consent for personal data processing (*n* = 1). A total of 290 patients participated in the study; 199 constituted the study group, and 91 the reference group. Baseline characteristics of enrolled patients are shown in Table 1. All individuals were Caucasian. Most participants (60%) were male and the mean age was 32.1 years. No statistically significant differences were observed between the study and reference group for demographic, clinical, and cognitive characteristics.

### 3.2. Conducted Cognitive Training Modules

In the study group, a total of 4746 cognitive training modules were begun, out of which 3395 (71.5%) were completed. On average, each patient completed 23 training modules, and started but did not finish 7 modules during the 12 months of the study. Table 2 summarizes the descriptive statistics of completed cognitive training modules in the study group. The number of patients who completed at least one cognitive training per month decreased from 102 in the first month of the study to approximately 50 in the last 4 months of the study. In these patients, the mean number of completed training modules each month remained stable throughout the study, and varied between 3.6 and 4.6 (out of maximum of 8 training modules that could potentially be completed per month).

### 3.3. Response Time Analysis

Each subsequent series of tasks was executed significantly faster than the previous one in both the study group (mean response time 666.5, 566.3 and 514.0 ms for 1st, 2nd and 3rd series, respectively, *p* < 0.001) and the reference group (mean response time 619.5, 526.7 and 514.4 ms for 1st, 2nd and 3rd series, respectively, *p* < 0.01). After 12 months of the study, patients from the study group needed significantly more time to respond to stimulus as compared to baseline (Table 3). In the reference group, response time also increased from baseline to the end of the study; however, in contrast to the study group, the difference was not statistically significant (Table 3). In general, the response time was not significantly longer in the study group than in the reference group (Table 4).

### 3.4. Correct Answers Rate

In the study group, the rate of correct answers increased markedly from 6.4% (T0) to 11.2% over the course of the study (T12, *p* = 0.0001, Table 3). The same pattern was observed for each of three series: the more training modules that were carried out, the more correct answers were given, with significant differences in the rate of correct answers between T0, T6, and T12 (*p* < 0.001 for each series). The rate of correct answers differed significantly between three series of the training: it was the highest for the 1st series (14.1%) and decreased gradually for the 2nd (9.8%) and 3rd series (7.9%, *p* < 0.001). ANOVA analysis including three series and three time points (T0, T6, T12) yielded significant main effects (*p* < 0.001) of both time and series.

In the reference group, the rate of correct answers increased from 3.8% to 8.9% but (in contrast to the study group) the difference was not deemed to be statistically significant (Table 3). The number of correct answers diminished for the subsequent three series (*p* < 0.001).

Comparing the rate of correct answers in the study group and the reference group (Table 4), patients from the study group had a significantly higher level of correct answers than individuals from the reference group (9.4% vs. 6.7%, *p* = 0.0004).

### 3.5. Incorrect Answers Rate

The rate of incorrect answers decreased slightly from T0 to T12 in both the study and the reference group (Table 3). In each series of the cognitive training, the number of incorrect answers increased (study group, 4.3%, 8.2%, 10.3% for 1st, 2nd, and 3rd series, respectively, *p* < 0.001, reference group, 5.4%, 8.4%, 9.9% for 1st, 2nd, and 3rd series, respectively, *p* < 0.001). No significant difference in the rate of wrong answers between the study group and the reference group was observed (Table 4).

### 3.6. Cognitive Fatigability

In both the study and the reference group, the indicator of fatigability (the number of lack of reactions) was greater for each subsequent series of tasks (*p* < 0.001 for both groups), and in each series it exceeded 80% of all responses given. Throughout the study, the fatigability level decreased significantly (by 3%) in the study group and remained unchanged in the reference group (Table 3). A significant difference in fatigability level was observed between the groups; the reference group demonstrated fatigability 2% higher than the study group (Table 4).

## 4. Discussion

A recently developed smartphone-based MONEO application offers a broad range of functionalities to remotely monitor clinical status, manage treatment, and improve the condition of individuals with schizophrenia [35]. Although no benefit in a long-term treatment compliance was shown [37], the usage of the application was associated with significant improvement in the clinical condition of the patients, along with decreased schizophrenia symptoms, as measured by the Calgary and PANSS scales [35]. The application was considered safe, with an adverse effect rate comparable to those described in other studies on smartphone-based tools [35]. Here, we used characteristics of patients’ performance in cognitive training exercises—correct and incorrect answers rate as well as cognitive fatigability—as a measure of cognitive functions. We showed that patients who had access to the full version of the MONEO application, and who were able to perform a cognitive training twice a week, displayed a meaningful progress in cognitive abilities. During this 1-year study, a statistically significant improvement in the rate of correct answers and cognitive fatigability, and a slight improvement in the rate of incorrect answers, were observed. In contrast, in patients supplied with an inactive version of the application, who performed cognitive training exercises only every 6 months, no significant changes in cognitive activities were demonstrated.

Consequently, after all mentioned above we may conclude that the MONEO application is a feasible tool to provide patients suffering from schizophrenia with an opportunity to conduct cognitive training exercises on a regular basis.

In the studies performed to date which investigate the use of mobile applications to conduct cognitive training exercises, the intervention has been short term, usually not longer than 1–2 months [38,39,40], however in one of the latest studies focused on cognitive rehabilitation of veterans with traumatic brain injury and posttraumatic stress disorder the interventions using mobile technology lasted for six months [41]. Furthermore, a possible occurrence of the ceiling effect, (preventing further progress after the first weeks of the training), is also hypothesized [39]. In contrast, in our much longer 12-month study, we observed constant progress in cognitive functions. The clinical condition of individuals with schizophrenia may fluctuate; therefore, (in contrast to the healthy population), to achieve good results in terms of cognitive rehabilitation, it seems that a cognitive stimulus must be of a long-term character. According to a couple of studies on mental health applications, a significant drop in mobile application use is observed after the first period of two weeks [38]; our study, however, does not confirm these observations. Although the number of training modules conducted decreased during the study, after 9 months it was still at a satisfactory level (reaching approximately half of the initial value) and remained stable to the end of the study. This might suggest that providing the patient with a complex platform (e.g., MONEO) that offers not only cognitive training capacity, but also tools for self-education and communication with professionals, can ensure more stable patient engagement.

Major limitations of this study are of methodological character. The patient compliance with all scheduled cognitive training modules was not complete. Therefore, the number of enrolled patients was larger than initially planned to assure an optimal number of observations. Still, the number of observations in the reference group was small, leading to non-significant conclusions. Another significant limitation of the study was the lack of measurement of cognitive functions apart from measurements made by the MONEO app. It is also possible that the improvement shown is the result of learning effect for the task. For this reason, no conclusion can be drawn about a significant improvement in cognitive function in patients, but only about the possibility of using CRT in the form of a mobile application in schizophrenia patients.

The results of this study show that cognitive training exercises performed on a regular basis for a longer period of time, facilitated by a smartphone-based application are feasible in schizophrenia patients. They may have the potential to significantly improve cognitive functioning of individuals with schizophrenia. Conclusions to be drawn from this feasibility study primarily focus on practicality of mentioned monitoring technology, yet not about its clinical efficacy or effectiveness. Therefore, correct answers and cognitive fatigability have potential to be functional markers of successful smartphone-based psychiatric rehabilitations in schizophrenia patients, but it needs further evaluation.

Given the growing popularity and ownership of remote devices, together with considerable efficacy and high acceptance levels of mobile solutions aimed at cognitive rehabilitation, further development of such self-administered tools is warranted. The development of technology in medicine leads to the creation of medical applications that use neural networks to analyze medical data. Data on patients’ cognitive functioning and their progress in cognitive training combined with clinical data on disease symptoms and drug compliance can be used to create records for artificial intelligence algorithms.

## Figures and Tables

**Figure 1 jcm-09-03681-f001:**
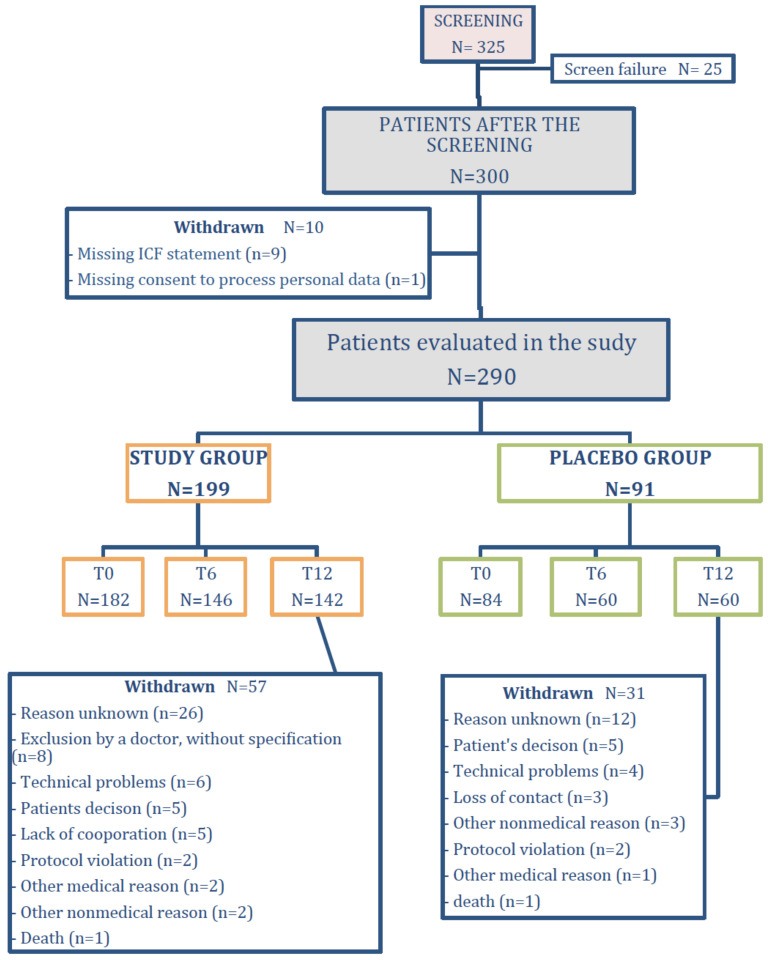
Participant flow throughout the study. Changes in the analyzed population during 12 months of the study are illustrated, based on the number of patients who underwent visits to the doctor’s office at baseline (T0), after 6 months (T6), and at the end of the study (after 12 months, T12).

**Table 1 jcm-09-03681-t001:** Baseline characteristics of the patients.

Variable	Study Group (*n* = 199)	Reference Group (*n* = 91)	*p*
Mean age, years	32.0 (5.92)	32.2 (6.94)	0.8
Sex—male	114 (57.3%)	60 (65.9%)	0.16
Race—Caucasian	199 (100%)	91 (100%)	1
Clinical status			
Total PANSS score	58.0 (20.3)	59.8 (23.7)	0.51
Calgary scale	4.0 (4.2)	3.4 (4.1)	0.26
CGI-S scale	2.7 (1.0)	2.7 (1.1)	1.0
Cognitive parameters			
Response time, ms	549.7 (131.5)	532.8 (160.7)	0.345
Correct answer rate, %	6.4 (4.7)	3.8 (4.1)	0.0001
Incorrect answer rate, %	6.4 (4.6)	4.7 (4.9)	0.0045
Fatigability, %	85.0 (5.6)	86.4 (6.6)	0.0631

Age and clinical scores are given as mean values with the standard deviation shown in parenthesis.

**Table 2 jcm-09-03681-t002:** Descriptive statistics of completed cognitive training modules in the study group.

Time Point	N	Mean (SD)	Median (IQR)	Min–Max
T1	102	4.6 (3.1)	4.0 (5.0)	1.0–17.0
T2	95	4.2 (2.8)	4.0 (5.5)	1.0–12.0
T3	89	3.8 (3.0)	3.0 (5.0)	1.0–18.0
T4	74	4.2 (3.1)	3.0 (4.0)	1.0–18.0
T5	69	3.8 (2.6)	3.0 (3.0)	1.0–10.0
T6	72	3.6 (2.8)	3.0 (4.0)	1.0–17.0
T7	62	4.4 (2.8)	4.0 (5.0)	1.0–12.0
T8	68	4.0 (3.2)	3.0 (6.0)	1.0–18.0
T9	46	4.1 (2.9)	3.5 (4.0)	1.0–15.0
T10	50	4.3 (3.0)	4.0 (3.8)	1.0–16.0
T11	55	4.0 (3.3)	3.0 (5.5)	1.0–18.0
T12	50	3.6 (2.9)	3.0 (4.0)	1.0–15.0
Overall	144	23.6 (29.5)	10.0 (30.0)	1.0–186.0

N, number of observations; SD, standard deviation; IQR, interquartile range; Min, minimum; Max, maximum; T1, in the first month of the study, etc.

**Table 3 jcm-09-03681-t003:** Changes in cognitive parameters within the study and reference group throughout the study.

Variable	Time	N	Mean (SD)	Median (IQR)	Min–Max	*p* (T0–T12)
Response time, ms						
Study group	T0	168	549.7 (131.5)	560.7 (188.4)	92.5–950.0	0.0001
	T12	63	627.2 (130.7)	667.6 (177.0)	279.6–840.6	
Reference group	T0	64	532.8 (160.7)	508.3 (213.2)	54.7–945.5	0.234
	T12	7	614.1 (156.4)	707.3 (210.4)	317.1–742.7	
Correct answers rate, %						
Study group	T0	168	6.4 (4.7)	5.8 (12.5)	0.0–19.8	0.0001
	T12	63	11.2 (5.8)	12.2 (16.7)	0.0–20.0	
Reference group	T0	64	3.8 (4.1)	1.8 (8.7)	0.0–14.2	0.088
	T12	7	8.9 (6.5)	8.0 (14.8)	1.6–16.4	
Incorrect answers rate, %						
Study group	T0	168	6.4 (4.6)	6.0 (7.2)	0.0–24.4	0.153
	T12	63	5.5 (4.1)	4.7 (5.6)	0.0–15.3	
Reference group	T0	64	4.7 (4.9)	2.7 (7.8)	0.0–17.6	0.643
	T12	7	4.1 (2.9)	4.4 (4.7)	0.7–8.9	

N, number of observations; SD, standard deviation; IQR, interquartile range; Min, minimum; Max, maximum; T0, study initiation; T12, end of study (after 12 months).

**Table 4 jcm-09-03681-t004:** Comparison of cognitive parameters between the study and reference group during 12 months of the study.

Variable	N	Mean (SD)	Median (IQR)	Min–Max	*p*
Response time, ms					
Study group	187	594.0 (133.1)	621.0 (194.3)	92.0–950.0	0.173
Reference group	66	565.5 (149.0)	591.1 (212.3)	186.3–809.2	
Correct answers rate, %					
Study group	187	9.4 (5.5)	9.3 (15.3)	0.0–20.0	0.0004
Reference group	66	6.7 (5.0)	6.2 (11.8)	0.5–15.1	
Incorrect answers rate, %					
Study group	187	6.3 (4.5)	5.6 (6.4)	0.0–35.3	0.131
Reference group	66	5.4 (4.0)	4.9 (6.1)	0.2–14.7	
Fatigability, %					
Study group	187	83.0 (5.3)	81.1 (6.4)	52.9–96.7	0.0183
Reference group	66	85.0 (6.0)	83.4 (10.7)	74.2–96.4	

N, number of observations; SD, standard deviation; IQR, interquartile range; Min, minimum; Max, maximum.

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
