# Peer review of "The Effect of Smartphone-Based Cognitive Training on the Functional/Cognitive Markers of Schizophrenia: A One-Year Randomized Study"

_jcm, 2020, doi:10.3390/jcm9113681_

Round 1

Reviewer 1 Report

The manuscripts is interesting, novelty and really useful for the implementation of ICT tools in our clinical practice.

The introduction section should be reformulated and some content of the discussion section should be moved to the introduction.

The structure of the introduction should included: cognition, remediation cognitive programs and the use of new ICT tools in the rehabilitation of cognitive performance. However the authors only include the cognition aspects and the second two points are describing in the discusssion section. I suggest to move the four first paragraphs of the discussions section to the introduction section.

The design of the study should be better explained. Which method of randomisation have been used to have 199 and 91 patients in each group? They have done 2 patients in the intervention group for 1 in the reference group?Please explain better.

In the method section the authors comment the use of the MONEO but they do not describe basic information about the application. This infromation is provided inthe discussion section, it should be moved. 

A flowchart with the information of the clinical trial should be added (participants in each group and each asssesment).

The results section is interesting and well-described.

The discussion section should be re-written. Some information should be moved to the introduction section and other to the method section. The discussion is not a discussion, the authors did not explain the results obtained and comprared with other researchers. This section should be modified and included the most important findings of the study and to compare them with other authors.

Author Response

Responses to the Reviewer 1

We thank the reviewer for his interest in our results and for giving him a positive evaluation of the manuscript. We tried to take advantage of his comments and improve the manuscript.

Point 1: The introduction section should be reformulated and some content of the discussion section should be moved to the introduction.

Answer 1: Major part of the Discussion section was moved to the Introduction section. The text (lines 241-274) has been moved to the introduction. In turn, the discussion was expanded:

“In the studies performed to date which investigate the use of mobile applications to conduct cognitive training exercises, the intervention has been short-term, usually not longer than 1-2 months [38,39,40], however in one of the latest studies focused on cognitive rehabilitation of veterans with traumatic brain injury and posttraumatic stress disorder the interventions using mobile technology lasted for six months [41].Furthermore, a possible occurrence of the ceiling effect, (preventing further progress after the first weeks of the training), is also hypothesized [39]. In contrast, in our much longer 12-month study, we observed constant progress in cognitive functions. The clinical condition of individuals with schizophrenia may fluctuate; therefore, (in contrast to the healthy population), to achieve good results in terms of cognitive rehabilitation, it seems that a cognitive stimulus must be of a long-term character. According to a couple of studies on mental health applications, a significant drop in mobile application use is observed after the first period of two weeks [38]; our study, however, does not confirm these observations. Although the number of training modules conducted decreased during the study, after 9 months it was still at a satisfactory level (reaching approximately half of the initial value) and remained stable to the end of the study. This might suggest that providing the patient with a complex platform (e.g. MONEO) that offers not only cognitive training capacity, but also tools for self-education and communication with professionals, can ensure more stable patient engagement.”

Point 2: The structure of the introduction should included: cognition, remediation cognitive programs and the use of new ICT tools in the rehabilitation of cognitive performance. However the authors only include the cognition aspects and the second two points are describing in the discusssion section. I suggest to move the four first paragraphs of the discussions section to the introduction section.

Answer 2: The structure of the Introduction section was rearranged according to the reviewer’s reservation. 

Point 3: The design of the study should be better explained. Which method of randomisation have been used to have 199 and 91 patients in each group? They have done 2 patients in the intervention group for 1 in the reference group? Please explain better.

Answer 3: The text explains the following: “The disproportion between the study group and the reference group was justified, first of all, by the need to gather a large study group to obtain significant results, on the other hand, we predicted a large number of patients who drop-out the study. " We have added now: "When designing the study, we assumed a 50% drop-out level of patients in the study group."

Point 4: In the method section the authors comment the use of the MONEO but they do not describe basic information about the application. This information is provided in the discussion section, it should be moved. 

Answer 4: We did not describe the application in detail as it would make the text very long. Detailed description of MONEO application functionality is provided by Krzystanek et al. (2018) [23]. In turn, in the present text we have described in detail the part of the application related to cognitive training.

Point 5: A flowchart with the information of the clinical trial should be added (participants in each group and each assessment).

Answer 5: We are grateful for that amendment. We added now the flowchart with the lacking information. Figure 1 presents now the flow of patients throughout the study: “Figure 1. Participant flow throughout the study. Changes in the analyzed population during 12 months of the study are illustrated, based on the number of patients who underwent visits to the doctor’s office at baseline (T0), after 6 months (T6), and at the end of the study (after 12 months, T12).”

Point 6: The discussion section should be re-written. Some information should be moved to the introduction section and other to the method section. The discussion is not a discussion, the authors did not explain the results obtained and comprared with other researchers. This section should be modified and included the most important findings of the study and to compare them with other authors.

Answer 6: In response to the reviewer’s reservation the following chapter was rewritten, to make the discussion more accurate:

“In the studies performed to date which investigate the use of mobile applications to conduct cognitive training exercises, the intervention has been short-term, usually not longer than 1-2 months [38,39,40], however in one of the latest studies focused on cognitive rehabilitation of veterans with traumatic brain injury and posttraumatic stress disorder the interventions using mobile technology lasted for six months [41].Furthermore, a possible occurrence of the ceiling effect, (preventing further progress after the first weeks of the training), is also hypothesized [39]. In contrast, in our much longer 12-month study, we observed constant progress in cognitive functions. The clinical condition of individuals with schizophrenia may fluctuate; therefore, (in contrast to the healthy population), to achieve good results in terms of cognitive rehabilitation, it seems that a cognitive stimulus must be of a long-term character. According to a couple of studies on mental health applications, a significant drop in mobile application use is observed after the first period of two weeks [38]; our study, however, does not confirm these observations. Although the number of training modules conducted decreased during the study, after 9 months it was still at a satisfactory level (reaching approximately half of the initial value) and remained stable to the end of the study. This might suggest that providing the patient with a complex platform (e.g. MONEO) that offers not only cognitive training capacity, but also tools for self-education and communication with professionals, can ensure more stable patient engagement.

Reviewer 2 Report

The manuscript is well presented and the research conducted within is scientifically sound. I have only one minor improvement for this manuscript. The introduction lacks an in depth rational, which was later mentioned in the discussion. Therefore, I would suggest to move Lines 241-274 into the Introduction, with minor editing. In Line 78 CT was not introduced. 

Author Response

Response to the Reviewer 2

Point 1: The manuscript is well presented and the research conducted within is scientifically sound. I have only one minor improvement for this manuscript. The introduction lacks an in depth rational, which was later mentioned in the discussion. Therefore, I would suggest to move Lines 241-274 into the Introduction, with minor editing. In Line 78 CT was not introduced.

We thank the reviewer for his interest in our results and for giving him a positive evaluation of the manuscript. We tried to take advantage of his comments and improve the manuscript.

Answer 1: The introduction has been rebuilt and lines 241-274 have been moved to the introduction as suggested by the reviewer. The discussion, in turn, was expanded, one paragraph was rewritten:

“In the studies performed to date which investigate the use of mobile applications to conduct cognitive training exercises, the intervention has been short-term, usually not longer than 1-2 months [38,39,40], however in one of the latest studies focused on cognitive rehabilitation of veterans with traumatic brain injury and posttraumatic stress disorder the interventions using mobile technology lasted for six months [41].Furthermore, a possible occurrence of the ceiling effect, (preventing further progress after the first weeks of the training), is also hypothesized [39]. In contrast, in our much longer 12-month study, we observed constant progress in cognitive functions. The clinical condition of individuals with schizophrenia may fluctuate; therefore, (in contrast to the healthy population), to achieve good results in terms of cognitive rehabilitation, it seems that a cognitive stimulus must be of a long-term character. According to a couple of studies on mental health applications, a significant drop in mobile application use is observed after the first period of two weeks [38]; our study, however, does not confirm these observations. Although the number of training modules conducted decreased during the study, after 9 months it was still at a satisfactory level (reaching approximately half of the initial value) and remained stable to the end of the study. This might suggest that providing the patient with a complex platform (e.g. MONEO) that offers not only cognitive training capacity, but also tools for self-education and communication with professionals, can ensure more stable patient engagement.”

CT (cognitive training) is explained in the text.
